# Heart Failure and Pancreas Exocrine Insufficiency: Pathophysiological Mechanisms and Clinical Point of View

**DOI:** 10.3390/jcm11144128

**Published:** 2022-07-15

**Authors:** Olivier C. Dams, Marlene A. T. Vijver, Charlotte L. van Veldhuisen, Robert C. Verdonk, Marc G. Besselink, Dirk J. van Veldhuisen

**Affiliations:** 1Department of Cardiology, University Medical Center Groningen, University of Groningen, 9700 RB Groningen, The Netherlands; m.a.t.vijver@umcg.nl (M.A.T.V.); d.j.van.veldhuisen@umcg.nl (D.J.v.V.); 2Department of Surgery, Amsterdam UMC, University of Amsterdam, 1100 DD Amsterdam, The Netherlands; c.l.vanveldhuisen@amsterdamumc.nl (C.L.v.V.); m.g.besselink@amsterdamumc.nl (M.G.B.); 3Amsterdam Gastroenterology Endocrinology Metabolism, 1100 DD Amsterdam, The Netherlands; 4Department of Gastroenterology and Hepatology, St. Antonius Hospital, 3435 CM Nieuwegein, The Netherlands; r.verdonk@antoniusziekenhuis.nl

**Keywords:** heart failure, interactions, pancreatic exocrine insufficiency, cardiac cachexia, congestion, malnutrition

## Abstract

Heart failure is associated with decreased tissue perfusion and increased venous congestion that may result in organ dysfunction. This dysfunction has been investigated extensively for many organs, but data regarding pancreatic (exocrine) dysfunction are scarce. In the present review we will discuss the available data on the mechanisms of pancreatic damage, how heart failure can lead to exocrine dysfunction, and its clinical consequences. We will show that heart failure causes significant impairment of pancreatic exocrine function, particularly in the elderly, which may exacerbate the clinical syndrome of heart failure. In addition, pancreatic exocrine insufficiency may lead to further deterioration of cardiovascular disease and heart failure, thus constituting a true vicious circle. We aim to provide insight into the pathophysiological mechanisms that constitute this reciprocal relation. Finally, novel treatment options for pancreatic dysfunction in heart failure are discussed.

## 1. Introduction

Heart failure is a complex clinical syndrome resulting from functional or structural disorders, leading to impaired ventricular filling or ejection of blood to the systemic circulation to meet metabolic requirements and accommodate venous return [1]. Heart failure is associated with decreased output (forward failure) and increased congestion (backward failure) that leads to disseminated organ dysfunction [2], which is generally associated with the severity of disease. Extensive work has been done in the field of dysfunction of the kidney [3,4], liver [5,6], the intestines and gut [7,8] and even the bone marrow [9], brain [10] and the placenta [11].

Pancreas function/dysfunction in heart failure has received little attention in the literature, which is somewhat surprising since malnutrition and gastrointestinal symptoms are common, as is dysfunctional insulin signaling [7,12,13]. The lack of research in this field may be related to the complexity of measuring pancreatic functioning and measuring the perfusion of human pancreatic tissue. Additionally, the pancreas has both an endocrine and exocrine function, and although there is significant interaction between the two systems, both are considered independent in functional testing.

Prior research has mostly been focused on the endocrine function of the pancreas in patients with heart failure, demonstrating impaired signaling and enhanced insulin clearance [14,15]. Heart failure is associated with an insulin resistant state, linked to the overactivation of the sympathetic nervous system [13,14]. As the endocrine pancreas represents the minority of the organ volume [16] and is relatively resistant to disturbed hemodynamics [17], the present review will focus on the exocrine function of the pancreas, its associated changes in heart failure, the pathophysiological mechanisms, and lastly the clinical consequences.

## 2. Pancreatic Circulation and Hemodynamics

The position of the pancreatic gland in the abdominal cavity and its complex vascular anatomy means that pancreatic blood flow is difficult to measure accurately using non-invasive methods [18]. Blood flow to the pancreas approximates about 1% of the cardiac output in healthy adults [19]. Most of the supply to the pancreas body and tail is derived from the splenic artery, a branch of the celiac trunk [19,20]. The superior and inferior pancreaticoduodenal arteries supply the head and neck of the pancreas. The superior pancreaticoduodenal artery branches from the gastroduodenal artery, which in turn branches from the common hepatic artery. The inferior pancreaticoduodenal artery is an early branch of the superior mesenteric artery. These two arteries thus represent an anastomosis between the celiac and mesenteric systems [20], which, similar to the cerebral circulation, allows for maintenance of pancreatic blood flow when vascular integrity is compromised. The venous drainage of the pancreas is entirely to the portal system through the splenic and superior mesenteric veins.

The healthy pancreas is capable of intrinsic regulation of blood flow by myogenic and metabolic regulatory mechanisms [21]. As a result of either reduced arterial pressure or elevated venous pressure, the pancreas increases capillary exchange capacity and is able to maintain oxygen extraction [21]. Additionally, reductions in arterial pressure stimulate autoregulation (e.g., decrease in vascular resistance) to allow for maintenance of sufficient blood flow [21]. Elevations in venous pressure are maintained solely by increased oxygen extraction, and lack of compensatory autoregulatory mechanisms. As has also been demonstrated in animal models, when venous (portal) pressures increase beyond specific thresholds, pancreatic blood flow is reduced [22].

The splanchnic veins are characterized by a much higher compliance and a large proportion of the circulating blood volume [23], hence they act as a venous reservoir [24]. Recently, modulation of this system has received attention as a treatment target in acute heart failure [25,26]. One of the hallmarks of the splanchnic vascular system, specifically the veins, is the large number of adrenergic receptors [27], allowing for a more pronounced vasomotor response. This allows for rapid fluid shifts to buffer changes in circulatory volume. The shift of blood from the splanchnic to the central compartment is capable of further increasing cardiac filling pressures and exacerbating decompensation. In case of expanded intravascular volume, the compliance of the splanchnic veins is decreased; when combined with pronounced venoconstriction due to strong adrenergic response, this renders the organs in the abdominal compartment, including the pancreas, prone to congestion (Figure 1B).

## 3. Pancreatic Structure and Function/Dysfunction

The pancreas is a dual-functional gland, performing both endocrine and exocrine functions. The endocrine functions are performed through the islets of Langerhans, which represent 1% of the pancreas and consist of hormone-secreting cells dispersed throughout and between the exocrine tissue. The islets are well-vascularized and secrete hormones, such as insulin, glucagon, somatostatin, ghrelin and pancreatic polypeptide [28], which act to maintain glucose homeostasis and regulate digestion. More than 95% of the pancreas is composed of exocrine tissue, consisting of acinar and ductal cells, which produce and discharge digestive enzymes and bicarbonate respectively into a system of intercalated ducts emptying into the proximal duodenum, allowing for digestion of lipids, carbohydrates and protein [29].

The regulation of exocrine pancreatic secretions involves a complex interplay of hormonal-hormonal and neurohormonal interactions [30]. The pancreas is extensively supplied with nerve fibers, nerve trunks and ganglia scattered throughout the tissue. The autonomic nervous system regulates pancreatic endocrine and exocrine secretions [31], especially the efferent parasympathetic pathways, consisting of the vagal central dorsal motor nucleus and pancreatic neurons, profoundly stimulate exocrine secretions [32]. The principal stimulatory hormones are cholecystokinin (CCK) and secretin; both of which are regulated by several feedback mechanisms [33]. Secretin, for example, is released primarily in response to increased acidity in the duodenum and stimulates the release of bicarbonate by ductal cells. CCK is released from endocrine cells in the gut by the presence of lipids and proteins, and acts through vagal afferents to stimulate the functional units of the exocrine pancreas—the acinar cells. 

### Pancreatic Exocrine Insufficiency

Pancreatic enzymes play a key role in the digestion of macronutrients and absorption of micronutrients. Due to a variety of (structural) diseases, pancreatic exocrine insufficiency may occur. The most common definition of pancreatic exocrine insufficiency is “a reduction of pancreatic exocrine activity in the intestine at a level that prevents normal digestion” [34]. The pathophysiology underlying the development of exocrine insufficiency is complex, and three distinct mechanisms can be implicated [35]:Injury to pancreatic exocrine parenchyma, resulting in reduced synthetic capacity (e.g., [recurrent] pancreatitis, cystic fibrosis, ageing)Reduced stimulation of pancreatic enzyme production (e.g., celiac disease, autonomic dysfunction)Pancreatic duct obstruction (e.g., malignancy)

In the case of generalized inflammation and malnutrition, exocrine secretions are decreased. Pancreatic exocrine insufficiency (PEI) is therefore seen in the wasting syndrome of, for example, chronic renal disease [36] and the critically ill [37]. It has previously been demonstrated that the presence of exocrine pancreatic insufficiency is associated with survival [38,39]. The main clinical consequence of (untreated) pancreatic exocrine insufficiency is maldigestion and subsequent malnutrition. In particular, the maldigestion of lipids may occur, resulting in deficiencies of fat-soluble vitamins (A, D, E, K) and minerals (e.g., copper, selenium, zinc) [34,40]. These deficiencies may cause downstream organ or immune dysfunction.

Pancreatic exocrine function is difficult to assess because of the inaccessibility of the gland’s secretions. Functional tests are either based on the measurement of secreted enzymes and bicarbonate (direct tests), or the investigation of secondary effects that are due to the lack of enzymes (indirect tests) (Table 1) [34]. Indirect tests are easy to administer, cheap, but generally less sensitive and particularly less-specific. The most employed test is the determination of fecal elastase-1 (FE-1), a pancreatic exocrine-specific enzyme that is not degraded in the bowel lumen [34]. This test requires a single stool collection, is easy to administer, cheap, and has demonstrated good diagnostic accuracy [41]. Alternatively, direct tests involve the collection of exocrine secretions through duodenal intubation [34]. These tests are more sensitive and specific, but their costs and invasive nature limits their routine use in clinical practice. Given the complexity of diagnosis, it is recommended that the evaluation of exocrine insufficiency is based on the assessment of the patient’s history and clinical state, weight-loss and nutritional status combined with functional testing [34].

In principle, pancreatic exocrine insufficiency is a manageable condition. The aim of treatment of exocrine insufficiency is to eliminate malabsorption and maldigestion and to normalize the nutritional state. The cornerstone of treatment is pancreatic enzyme replacement therapy (PERT) [34]. PERT involves oral administration of enzyme preparations and has been shown to improve outcomes [42,43,44] in both malignant and benign conditions associated with exocrine dysfunction.

## 4. Studies on Pancreatic Injury in Heart Failure

In heart failure, a combination of contributors, such as low cardiac output, activation of the renin-angiotensin-aldosterone system, and natriuretic peptide axes, and the sympatho-sympathetic reflex lead to tissue injury mainly through hypoperfusion and/ or congestion [1]. Herein we summarize the few prior studies on pancreatic involvement in animal and human models of heart failure and cardiac dysfunction.

Acute pancreatic injury has been well described in earlier studies in the setting of cardiac surgery [45,46,47,48,49,50,51]. In this population, post-operative biochemical derangement (hyperamylasemia/hyperlipasemia) is prevalent, detected in nearly one third of patients [47]. Additionally, hemodynamic factors, such as hypotension, low cardiac output, and increased bypass time, are independent predictors [47]. Although the presentation is usually subtle, severe (fulminating) pancreatitis has been documented in some patients [45,46,48,51,52]. The development of ischemic pancreatitis is associated with microvascular impairment and stasis, formation of thrombi, release of digestive enzymes and inflammation [53,54]. In severe cases, this proceeds to systemic inflammation leading to systemic inflammatory response syndrome and multi-organ involvement, an infrequent but well-recognized, severe, complication of cardiac surgery, termed post-pump pancreatitis.

In (acute) heart failure induced by rapid ventricular pacing in dogs, pancreatic perfusion declines rapidly, even before significant limitations in renal blood flow are seen [55]. Older studies in humans already demonstrated that in cases of acute cardiogenic shock, the incidence of pancreatic injury is 50–55% in those with acute tubular necrosis and 9% in those without [56]. The disproportional decline in pancreatic perfusion is also confirmed by an earlier animal study [57], utilizing a pig-model of cardiogenic shock induced by pericardial tamponade. This study showed a rapid, highly disproportional, reduction in pancreatic perfusion and increase in vascular resistance. Interestingly, this disproportionality is partially reversible with pharmacological renin-angiotensin inhibition [57]. Histopathological follow-up analyses of pancreatic tissue in the abovementioned dog model demonstrated atrophy of the acinar cells specifically [58,59], indicating that an association between the hemodynamic perturbations of heart failure and loss of exocrine parenchyma exists.

The interaction between the abdominal compartment and central hemodynamics, inflammation, malnutrition and catabolism has been established previously in several studies in patients with cardiac cachexia. Gut hormones, such as leptin and ghrelin, are elevated in cachectic patients [60,61], a finding that is thought to reflect the anabolic/catabolic imbalance. A recent prospective study in an advanced heart failure population demonstrated that the endocrine peptide somatostatin, expressed in the stomach, small intestine and pancreas, and a potent inhibitor of pancreatic hormones and exocrine secretions, is elevated in chronic heart failure and independently related to right-sided filling pressures and cardiac index [62]. Cardiac cachexia itself is also strongly associated with right-sided filling pressures [7]; the elevated intestinal hormones, such as somatostatin, could thus reflect the pathologic hormonal milieu, but also the effect of congestion on the abdominal compartment.

Only three relatively small studies evaluated pancreatic function in the human heart failure population [63,64,65]. The largest, most recent, study examined FE-1 in 104 patients with mostly mild heart failure [65] and showed that 56% of these patients had evidence for exocrine insufficiency (FE-1 < 200 µg/g), which appeared to be associated with both gastrointestinal symptoms and the severity of heart failure. The high prevalence of pancreatic exocrine insufficiency was also found in an earlier Turkish report [64]. This study, in 52 patients with mild and moderate heart failure and 31 healthy controls, also showed a high prevalence of pancreatic exocrine insufficiency (70%) in severe heart failure, and strong associations between FE-1 and cachexia/malnourishment and exocrine insufficiency. In a third study, Vujasinovic et al. [63] provided evidence for decreased circulating fat-soluble vitamins/micronutrients, particularly vitamin D, in all patients with heart failure and pancreatic exocrine insufficiency. Although these three studies are important, only little information on patient selection, baseline characteristics, and etiology was provided, and it appears that a relatively mild heart failure population was examined, based on natriuretic peptides and echocardiography. Nonetheless, these studies support the concept that heart failure is associated with pancreatic exocrine insufficiency.

## 5. Mechanisms of Pancreatic Damage in Heart Failure

Almost a century ago, it was first hypothesized that repetitive attacks of acute pancreatitis progressed to chronic pancreatitis [66]. The hallmark of chronic pancreatitis is the progressive, fibrotic destruction of tissue, resulting from a combination of (genetic) susceptibility and incremental damage from a repetitive injury. In contrast to other organs, such as the skin, liver and intestines, but similar to the heart, the adult pancreas displays a limited capacity to regenerate [67,68] and a tendency towards fibrogenesis and cellular atrophy in response to injury. Although acinar cells increase their division rate after injury [69,70], tissue mass is only partially restored, allowing for the development of exocrine insufficiency. The suggested mechanisms of pancreatic dysfunction resulting in exocrine insufficiency in heart failure are discussed in this chapter and depicted in Figure 2.

### 5.1. Ageing

Across the human lifespan, pancreatic structure and function changes: volume decreases, ducts dilate, and exocrine function gradually decreases [71]. Similar to other abdominal organs, such as the liver [72], there is an age-dependent decline in perfusion [73], which contributes to gradual atrophy of acinar cells and eventually (sub-clinical) exocrine insufficiency. This process appears to develop independent of co-morbidities. Prior studies in healthy individuals showed that age correlates with exocrine secretions and more than one fifth of healthy older adults exhibit pancreatic exocrine insufficiency [74,75]. It can be hypothesized that repetitive hits of accumulating hypoperfusion in heart failure accelerate the changes seen in ageing, putting patients at increasing risk for (premature) development of exocrine dysfunction (Figure 1C).

### 5.2. Hemodynamics

The pancreas is vulnerable to ischemic injury, a factor that plays an important role in the development of a wide range of pancreatic disease, particularly pancreatitis [76,77,78]. Ischemic injury occurs rapidly with subsequent induction of a significant inflammatory response and acinar cell injury [77,79]. The functional consequences of temporary ischemic damage on the exocrine pancreas are unknown, as studies on tissue regeneration after recovery from ischemic injury humans are scarce [29]. Older data from autopsy studies on patients with ischemic pancreatic injury [59] demonstrated that a wide range of histopathological changes occur, especially in the acini. Peripheral acinar atrophy, as already described more than 35 years ago by Takahashi et al., is found exclusively in congestive heart failure, with more than 10% of patients exhibiting this finding at autopsy [59]. Similar histological changes are seen in models of ischemia related to atherosclerosis in ageing or long-standing hypoperfusion [71,77]. These data suggest that chronic ischemic injury damages the functional unit of the exocrine pancreas - the acinus.

Under conditions of acute heart failure manifesting as cardiogenic shock, there is systemic vasoconstriction to maintain effective circulation. This response is disproportionate in the mesenteric circulation. Although the sympathetic nervous system mediates vasoconstriction, the predominant cause of this disproportionate response in the mesenteric circulation seems to be the renin-angiotensin axis [57,80,81]. The mesenteric vascular smooth muscle has a disproportionate affinity for angiotensin II [82]. A local renin-angiotensin system is also present in the pancreas, a feature that can further amplify oxidative stress and tissue injury [83,84]. This renders the pancreas susceptible to injury in heart failure, where renin-angiotensin overactivation is a central feature underlying the pathophysiology.

In (advanced) heart failure, the abdominal compartment is subjected to progressive and persistent volume overload, a factor that compromises oxygen supply and independently predicts outcome and the development of cardiac cachexia [5,7,12]. Fluid overload or high (right) ventricular filling pressures, and elevated central venous pressure, are transmitted directly to the hepatic veins and sinusoidal beds in the liver [85]. This results in sinusoidal congestion and peri-sinusoidal edema, decreasing the oxygen diffusion of hepatocytes. Due to their close interaction and coupling, where venous drainage of the pancreas occurs entirely through the portal system, the presence of acute liver injury itself puts patients at risk of pancreatic injury. Portal hypertension has been shown to cause fibrosis and damage the intima of the pancreatic vein wall, resulting in impaired drainage of blood flow [86,87]. Given its anatomical position and complex auto-regulation [22], we hypothesize congestive pancreatopathy is a common, unrecognized, clinical entity in patients with heart failure.

### 5.3. Inflammation

Inflammatory processes of the pancreas are a well-known clinical entity [88]. Acute pancreatitis is driven by a local and systemic inflammatory cascade, resulting in tissue atrophy and fibrosis [89,90]. The disease seems to be triggered by inappropriate activation of pancreatic enzymes and subsequent auto-digestion [91]. Damage is enhanced by a sterile inflammatory response consisting of the activation of transcription factors, such as NF-κB [90,92], cytokine dysregulation [93,94] and eventually recruitment of macrophages and T cells [95,96]. It is unknown whether a single attack of acute pancreatitis has significant (long-term) effects on exocrine function. Animal studies demonstrated that exocrine function is decreased during acute pancreatitis [97,98,99]. Studies in humans showed a high prevalence of exocrine insufficiency after recovery from acute pancreatitis (27.1%), with a higher prevalence in more severe disease [100,101].

Persistent (low-grade) sterile inflammation also represents an important pathogenic mechanism in heart failure, which seems to be unresponsive to treatment [102,103] and correlates with severity of disease [104] and the development of cardiac cachexia. Elevated inflammatory biomarkers are a hallmark feature of heart failure and thought to be both cause and consequence of disease progression. Additionally, several other associated conditions, such as atrial fibrillation [105], coronary artery disease [106], and anemia [107], are to some extent driven by inflammation. This inflammatory state is associated with end-organ dysfunction through direct tissue damage, impairment of extracellular matrix networks, and alterations in metabolism and endocrine signaling [108,109]. It is hypothesized that chronic (low-grade) inflammation in heart failure facilitates damage to the exocrine pancreas parenchyma. 

### 5.4. Autonomic Dysfunction

Heart failure is characterized by marked autonomic dysfunction, i.e., increased sympathetic activation (and circulating catecholamines) on the one hand, and reduced vagal (or parasympathetic) activity on the other [110,111]. This disturbed autonomic balance can be (partly) corrected by drug treatment [112]. Interestingly, baroreflex sensitivity at baseline is reduced in patients with heart failure but is unaffected by unloading the cardiopulmonary baroreceptors (through lower body negative pressure application) [113], suggesting that chronic overstimulation leads to insensitivity of the autonomic nervous system.

The autonomic nervous system also plays a role in the regulation of pancreatic exocrine and endocrine function and the secretion of enzymes [28]. Indeed, cholecystokinin stimulates pancreatic exocrine secretion through vagal neurons, and so it is conceivable that in heart failure a chronic reduction of parasympathetic activation and overstimulation of the sympathetic nervous system may lead to insensitivity of the entero-pancreatic reflexes, similar to what is described above, thus causing pancreatic exocrine insufficiency [113,114]. Interestingly, in patients with diabetes mellitus, a condition that is also common in patients with heart failure, reduced exocrine stimulation is also observed, which may be the result of reduced parasympathetic activity [115]. Studies will be needed to investigate the effects on vagal stimulation on pancreatic enzyme secretion and function, in patients with heart failure versus those without, to investigate whether the pancreas in heart failure may also be less responsive to vagal stimulation.

## 6. Summary and Clinical Implications

The aim of this review is to focus on the exocrine function of the pancreas, its associated changes in heart failure and the pathophysiological mechanisms. Although only three previous reports have directly addressed the relationship between heart failure and exocrine insufficiency using direct clinical markers (e.g., FE-1), the various studies discussed in this review showed that several of the hallmarks and consequences of heart failure, such as abnormal hemodynamics and repetitive tissue injury, autonomic dysfunction and systemic inflammation, provide a detrimental environment capable of damaging specifically pancreatic exocrine tissue. A two-hit model in which (1) ongoing/chronic hemodynamic injury and inflammation leading to acinar atrophy and (2) impaired tissue stimulation subsequent to autonomic dysfunction is proposed (Figure 1A). It is hypothesized that this condition leads to further clinical deterioration by accelerating wasting and malnutrition. Although it is reasonable that these pathological mechanisms are capable of disproportionally damaging pancreatic tissue, the development of pancreatic insufficiency as heart failure worsens is still unclear. Based on prior data from studies on organ dysfunction in heart failure, one can assume a progressive trend related to heart failure severity, but this is certainly partly hypothetical. Future (longitudinal) studies in both mild and advanced heart failure populations are therefore encouraged.

Patients with severe (advanced) heart failure may develop cardiac cachexia, which is a strong independent prognostic factor [116] that intensifies the pathological cardiac remodeling process [117]. The pathogenesis of cardiac cachexia is complex and involves systemic dysregulation of inflammatory and metabolic pathways. Pancreatic exocrine insufficiency may be both a contributor and a consequence of cardiac cachexia. Several features of cardiac cachexia, such as abdominal congestion, inflammation and sympathetic overactivation [7,12,118,119], overlap with potential mechanisms of pancreatic injury. Alternatively, pancreatic exocrine insufficiency can also (further) exacerbate cardiac cachexia. Muscle wasting and (protein) malnutrition, for example, are common in heart failure [120], but could also be related to pancreatic exocrine insufficiency. This insufficiency may be treatable, thus making it a potential element of heart failure and cardiac cachexia that is amenable to intervention. Larger observational studies addressing the prevalence of pancreas exocrine insufficiency in patients with heart failure are needed to determine which patients are affected most. We encourage other researchers to identify patient characteristics and risk factors for developing pancreatic disease, therewith possibly providing new options for prevention or treatment. Finally, supplemental therapies (PERT) could be further explored in patients with pancreas dysfunction and heart failure.

In addition to aggravating the wasting of advanced heart failure, pancreatic exocrine insufficiency may result in micronutrient deficiencies subsequent to maldigestion of lipids. In particular, deficiencies in several fat-soluble minerals, such as selenium and zinc, may occur [121]. These micronutrients are essential for cardiac mitochondrial function and oxidative stress response, and their presence is associated with positive outcomes in heart failure [122]. Furthermore, reduced circulating fat-soluble vitamins may further deteriorate the clinical status in heart failure. Vitamin D deficiency, for example, aggravates the wasting of cardiac cachexia by compromising bone mineral density. Additionally, vitamin D plays a significant role in cardiovascular homeostasis and remodeling, and prior studies showed that vitamin D deficiency is associated with cardiovascular events and negative outcomes in heart failure [123,124,125,126,127]. Studies that tested supplementation of vitamin D in heart failure have shown that this treatment reduces renal activity [128], but it does not improve outcomes in heart failure [129]. The exact reasons why this treatment is not beneficial over the long-term is unknown [130]. Interestingly, vitamin D supplementation also does not affect bone turnover [131], a finding that could be partially explained by dysfunctional vitamin D handling subsequent to (persistent) exocrine pancreatic dysfunction. Lastly, deficiencies of vitamins A, E and K reduce physical functioning by causing visual impairment, neurological symptoms and coagulopathy [40]. We encourage future trials that investigate supplementation of these minerals and vitamins in heart failure to also consider exocrine pancreatic function.

## 7. Conclusions

Although extensive work has been done on organ dysfunction in heart failure, the pancreas remains largely understudied. This review presents the pathophysiological mechanisms and available data that together suggest heart failure is associated with pancreatic exocrine insufficiency, a feature that can further deteriorate clinical status through malnutrition and maldigestion. As pancreatic exocrine insufficiency is a treatable, largely unrecognized condition, further research evaluating its prevalence and treatment in heart failure is encouraged.

## Figures and Tables

**Figure 1 jcm-11-04128-f001:**
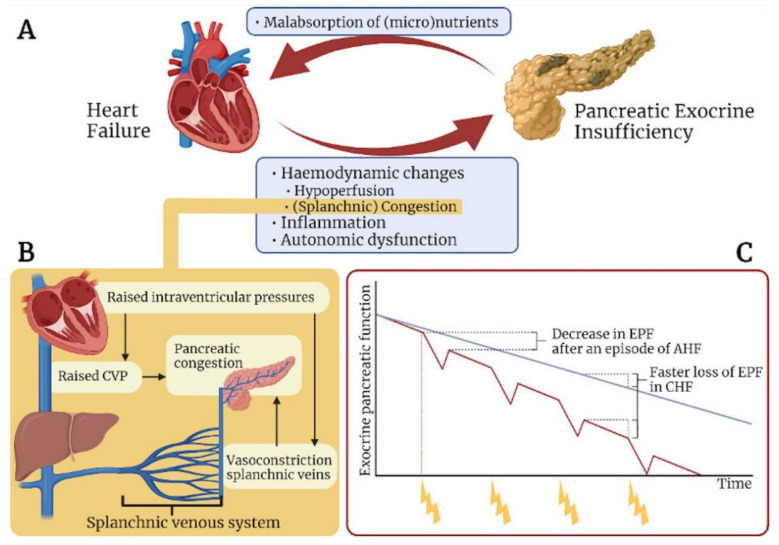
(**A**) Possible interactions between heart and pancreas. Several factors are hypothesized to affect the pancreatic exo-crine tissue in heart failure. This is mainly due to ischemic injury and congestion, leading to necrosis of acinar cells. As a result, EPF is suggested to decrease, followed by malabsorption of nutrients, which further deteriorates heart failure. (**B**) Schematic view of the venous system affected by heart failure. Due to decreased organ perfusion, the venous splanchnic system contracts, aiming to increase the circulatory volume. Together with raised CVP, these mecha-nisms are believed to lead to compromised venous drainage of the pancreas, resulting in congestion. (**C**) Graph showing the proposed loss of EPF in the normal situation compared to patients with heart failure. Every episode of AHF can be seen as a new attack on the pancreatic exocrine tissue. Repetitive hits, in combination with chronic heart failure, is hypothesized to result in exocrine pancreatic insufficiency.

**Figure 2 jcm-11-04128-f002:**
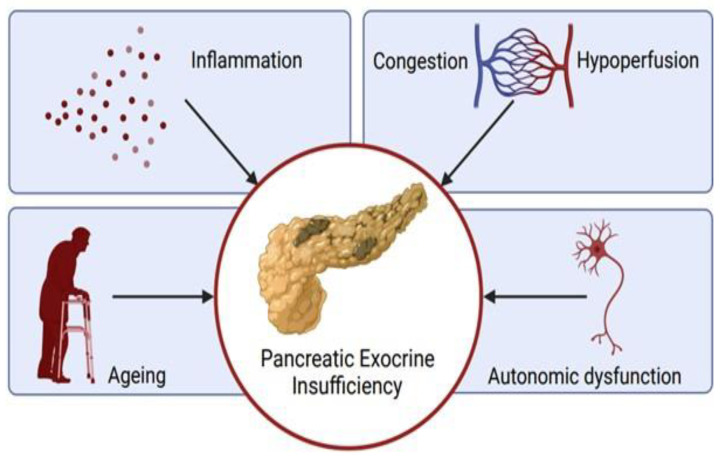
Proposed mechanisms of pancreatic damage in heart failure. Several factors have been identified that possibly damaged the exocrine pancreatic tissue in patients with heart failure. We know that in healthy individuals, pancreatic exocrine insufficiency can develop over time. In patients with heart failure, the pancreas is supposedly affected by deranged hemodynamics (e.g., congestion and hypoperfusion), chronic low-grade inflammation and autonomic dysfunction, therefore possibly accelerating the development of pancreatic exocrine insufficiency. Figure created with BioRender.com (accessed on 13 May 2022).

**Table 1 jcm-11-04128-t001:** Diagnostic tools of measuring pancreatic exocrine insufficiency.

Test	Advantages	Disadvantages
	Non-invasive	
**Fecal elastase-1 (FE-1)** single stool measurement	Easy to determine Low patient burden Cheap	Low accuracy in mild to moderate PEI Frequent false positives Not accurate in case of diarrhea
**Coefficient of fat absorption (CFA)** 3-day stool collection	High accuracy, former gold-standard	Requires 3-day stool collection and 5-day strict diet
**13C-mixed triglyceride breath test**	Can be used for treatment monitoring	Not completely validated and no agreement on protocol Limited availability Time consuming (6 h)
**Trial of PERT** Pancreatic Enzyme Replacement Therapy	Also possible treatment Low patient burden	Only in symptomatic patients, clinically highly suspected Risk of over- and under diagnosis
	Invasive	
**CCK-stimulated pancreatic function test** measuring enzyme secretion acinar cells in duodenum	Most direct measure of enzyme secretion Accuracy for detecting mild/moderate insufficiency	Duodenal intubation Long procedure (2 h) Expensive High patient burden
**Secretin stimulated pancreatic function test** measuring HCO_3_^−^ and fluid secretion ductal cells in duodenum	Most direct measure of HCO_3_^-^ and fluid secretion Accuracy for detecting mild/moderate insufficiency	Duodenal intubation Long procedure (2 h) Expensive High patient burden
**Secretin—CCK test**, combination of the above		
	Imaging	
**Endoscopic retrograde cholangiopancreatography** **Computed tomography** **Endoscopic ultrasound** **Magnetic resonance imaging**	Identify structural abnormalities (ductal changes, hyperechoic regions, cysts, parenchymal lobularity, calcifications)	Poor sensitivity for mild disease Evaluates probability of PEI in chronic pancreatitis

CCK—cholecystokinin.

## Data Availability

Not applicable.

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
