# Peer review of "Heart Failure and Pancreas Exocrine Insufficiency: Pathophysiological Mechanisms and Clinical Point of View"

_jcm, 2022, doi:10.3390/jcm11144128_

Round 1

Reviewer 1 Report

The manuscript is well written, in a very comrehensive manner. It provides detailed overview of the mechanism of pancreatic damage in the heart failure. Although the influence of heart failure and consequent venous congestion on the function of different organs has been widely studied, the data about the influence on pancreatic function are scarce. Therefore, this illustrative review will provide new insight into pathophisiology of pancreatic damage in heart failure, and treatment options.

Reviewer 2 Report

This review article is not convincing, as only three previous reports have addressed the relationship between the heart failure and pancreatic exocrine insufficiency.

1.      The authors mixed acute and chronic heart failure. It is reasonable that acute heart failure provides dramatic impacts on organ hemodynamics, but it is not clear in this paper whether pancreatic exocrine insufficiency progresses as chronic heart failure worsens or not.

2.      The suggested mechanisms are common for all organs and not specific for the pancreas. Is this topic really suitable for a review article?

Reviewer 3 Report

The manuscript titled "Heart Failure and Pancreas Exocrine Insufficiency: Pathophysiological Mechanisms and Clinical Point of View " is well written, provides a well organized and comprehensively review.

I just have one comment: The section on clinical implications and future trials is too short. Please comment on what should be the next research steps (e. g. heart failure - atherosclerosis - inflammation - hypoxia - chronic pancreatitis -  pancreas exocrine insufficiency).

Round 2

Reviewer 2 Report

The authors have responded seriously to the questions raised by the reviewer, and have added the limitations of this review article.